# The Anticancer Ruthenium Compound BOLD-100 Targets Glycolysis and Generates a Metabolic Vulnerability towards Glucose Deprivation

**DOI:** 10.3390/pharmaceutics14020238

**Published:** 2022-01-20

**Authors:** Dina Baier, Beatrix Schoenhacker-Alte, Mate Rusz, Christine Pirker, Thomas Mohr, Theresa Mendrina, Dominik Kirchhofer, Samuel M. Meier-Menches, Katharina Hohenwallner, Martin Schaier, Evelyn Rampler, Gunda Koellensperger, Petra Heffeter, Bernhard Keppler, Walter Berger

**Affiliations:** 1Institute of Inorganic Chemistry, University of Vienna, 1090 Vienna, Austria; dina.baier@univie.ac.at (D.B.); beatrix.schoenhacker-alte@gmx.at (B.S.-A.); mate.rusz@univie.ac.at (M.R.); theresa.mendrina@univie.ac.at (T.M.); katharina.hohenwallner@univie.ac.at (K.H.); martin.schaier@univie.ac.at (M.S.); evelyn.rampler@univie.ac.at (E.R.); bernhard.keppler@univie.ac.at (B.K.); 2Center for Cancer Research and Comprehensive Cancer Center, Department of Medicine I, Medical University Vienna, 1090 Vienna, Austria; christine.pirker@meduniwien.ac.at (C.P.); thomas.mohr@mohrkeg.co.at (T.M.); dominik.kirchhofer@meduniwien.ac.at (D.K.); petra.heffeter@meduniwien.ac.at (P.H.); 3Research Cluster “Translational Cancer Therapy Research”, 1090 Vienna, Austria; 4Department of Analytical Chemistry, Faculty of Chemistry, University of Vienna, 1090 Vienna, Austria; samuel.meier@univie.ac.at (S.M.M.-M.); gunda.koellensperger@univie.ac.at (G.K.)

**Keywords:** BOLD-100/KP1339, ruthenium, chemotherapy resistance, 2-deoxy-d-glucose, glycolysis, ER stress, autophagy, lysosome, colon cancer, pancreatic cancer

## Abstract

Cellular energy metabolism is reprogrammed in cancer to fuel proliferation. In oncological therapy, treatment resistance remains an obstacle and is frequently linked to metabolic perturbations. Identifying metabolic changes as vulnerabilities opens up novel approaches for the prevention or targeting of acquired therapy resistance. Insights into metabolic alterations underlying ruthenium-based chemotherapy resistance remain widely elusive. In this study, colon cancer HCT116 and pancreatic cancer Capan-1 cells were selected for resistance against the clinically evaluated ruthenium complex sodium trans-[tetrachlorobis(1H-indazole)ruthenate(III)] (BOLD-100). Gene expression profiling identified transcriptional deregulation of carbohydrate metabolism as a response to BOLD-100 and in resistance against the drug. Mechanistically, acquired BOLD-100 resistance is linked to elevated glucose uptake and an increased lysosomal compartment, based on a defect in downstream autophagy execution. Congruently, metabolomics suggested stronger glycolytic activity, in agreement with the distinct hypersensitivity of BOLD-100-resistant cells to 2-deoxy-d-glucose (2-DG). In resistant cells, 2-DG induced stronger metabolic perturbations associated with ER stress induction and cytoplasmic lysosome deregulation. The combination with 2-DG enhanced BOLD-100 activity against HCT116 and Capan-1 cells and reverted acquired BOLD-100 resistance by synergistic cell death induction and autophagy disturbance. This newly identified enhanced glycolytic activity as a metabolic vulnerability in BOLD-100 resistance suggests the targeting of glycolysis as a promising strategy to support BOLD-100 anticancer activity.

## 1. Introduction

The reprogramming of cellular energy metabolism characterizes a hallmark of diverse types of cancer to efficiently fuel biomass production and sustain cell proliferation [1]. The mechanisms underlying a metabolic reprogramming are manifold and highly complex. Besides other metabolic changes, including alterations in oxidative phosphorylation or glutaminolysis [2], diverse types of solid cancers show enhanced glycolysis to convert glucose to lactate, even under aerobic conditions, a phenomenon termed the “Warburg effect” [3,4]. In the development of anticancer therapy resistance, cellular metabolism, for example, glycolytic activity, is often further enhanced [5]. In this regard, several combined or independently emerging mechanisms have been identified, including (1) the upregulated expression of (rate-limiting) enzymes of the glycolytic pathway [2,6], such as hexokinase 2 (HK2) and phosphofructokinases (PFK), (2) enhanced expression of glucose transporters (GLUT) [7,8,9], (3) aberrant activity of signaling pathways sensing the metabolic state [10,11], and (4) Myc-induced metabolic reprogramming [12,13]. Accordingly, enhanced cellular glycolytic activity confers a poor prognosis in patients suffering from diverse types of solid tumors [5,7,8]. This increased and vital glycolytic activity in (therapy-resistant) cancer cells translates into an exploitable vulnerability [14] and, hence, provides a rational therapy target.

The glucose analogue 2-deoxy-d-glucose (2-DG) (Appendix A) is irreversibly converted to 2-DG-6-phosphate by HK in the first step of glycolysis and cannot be further metabolized, thereby halting glycolysis [15] and culminating, depending on glycolysis dependency, in growth inhibition and cell death by diverse mechanisms of action [16,17,18]. Kurtoglu et al. showed that glycolysis inhibition by 2-DG affected protein *N*-glycosylation, causing unfolded protein response (UPR) induction via upregulation of the endoplasmic reticulum (ER) chaperone glucose-regulated protein of 78 kDa (GRP78), and finally resulting in apoptosis [19]. Hence, glucose deprivation upregulates GRP78 expression at the transcriptional [20] and translational [21] levels. Furthermore, 2-DG-induced ER stress and ATP-depletion can prompt autophagy [22,23], a cellular catabolic organelle recycling program. Under basal conditions, autophagy ensures cellular survival by the removal of damaged proteins or organelles. If hyperactivated, autophagy can lead to a widely caspase-independent form of programmed cell death [24,25]. Other described modes of action concerning 2-DG were associated with perturbations of cell cycle progression and antiproliferative activity [19]. 2-DG was investigated in clinical trials as a single agent [26,27] or combined with the chemotherapeutic docetaxel for the treatment of advanced solid tumors [28]. Overall, 2-DG therapy was well-tolerated and patients experienced mild side effects, including nausea and fatigue. Concerning the clinical combination approach, the authors emphasized that 2-DG should be combined with chemotherapeutic agents to yield a more significant therapeutic success. This is also supported by several in vitro experiments that demonstrated an enhanced anticancer activity of chemotherapy upon combination with 2-DG [29]. Moreover, glucose deprivation with 2-DG combined with platinum-based chemotherapy was shown to overcome acquired metal drug resistance [11,30].

Insights into the mechanisms in response to ruthenium-based anticancer therapy and underlying an acquired resistance against this approach, especially concerning metabolic alterations, remain widely enigmatic. In a recent publication, we demonstrated that metabolite abundances and metabolic flux alterations underlying acquired platinum- and ruthenium-based chemotherapy differ distinctly [31]. The aim of the present study was to investigate the molecular mechanisms characterizing acquired resistance against the clinically evaluated ruthenium complex BOLD-100 (sodium trans-[tetrachlorobis(1H-indazole)ruthenate(III)] with cesium as an intermediate salt form (Appendix A); predecessor molecules: IT-139/NKP-1339/KP1339). BOLD-100 is an inhibitor of stress-induced GRP78 upregulation, thus disrupting ER homeostasis and inducing ER stress and UPR. This is reflected by the phosphorylation of eukaryotic translation initiation factor 2A (eIF2A) [32] and caspase-8-dependent cell death [33]. After the successful completion of a phase I safety and activity study [34], BOLD-100 is currently in phase Ib clinical investigation, in combination with folinic acid, fluorouracil, and oxaliplatin (FOLFOX regimen), for the treatment of colorectal, pancreatic, and gastric cancers, as well as cholangiocarcinoma (NCT04421820). Besides BOLD-100, imidazolium [trans-tetrachloro(1H-imidazole)(S-dimethylsulfoxide)ruthenate(III)] (NAMI-A) was clinically investigated as the first ruthenium compound reaching phase II study stage but failed, due to an insufficient activity, a low response rate, and the development of painful blisters at higher doses [35,36]. The BOLD-100 progenitor indazolium [trans-tetrachlorobis(1H-indazole)ruthenate(III)] (KP1019) followed NAMI-A into clinical investigation [36]. Due to an insufficient solubility that prevented the identification of a maximum tolerated dose and also based on the low toxicity profile, KP1019 was replaced by the more soluble sodium salt analogue BOLD-100 [37,38].

The development of (chemo)therapy resistance presents a major obstacle in anticancer therapy. Therefore, it is essential to dissect the mechanisms underlying acquired BOLD-100 resistance concomitantly to early clinical development. Hence, in the present study, colon cancer HCT116 and pancreatic cancer Capan-1 cells were selected for acquired BOLD-100 resistance and investigated for the underlying molecular mechanisms by multi-omics approaches and cell/molecular biological and analytical chemistry methods. Summarizing, we identified specific metabolic changes upon short- and long-term BOLD-100 treatment associated with altered glycolytic activities. Accordingly, we show that these metabolic adaptations contribute to acquired BOLD-100 resistance and open novel approaches for synergistic pharmacological interventions.

## 2. Materials and Methods

### 2.1. Chemicals and Drugs

BOLD-100 was obtained from Bold Therapeutics Inc. (Vancouver, Canada) and dissolved in dimethyl sulfoxide (DMSO) to generate a 20 mM stock. 2-DG was purchased from Sigma-Aldrich Inc. (St. Louis, MO, USA) and freshly dissolved in cell culture medium prior to use. Bis-2-[5-(phenylacetamido)-1,3,4-thiadiazol-2-yl]ethyl sulfide (BPTES) and chloroquine (CQ) were obtained from Selleckchem (EUBIO, ANDREAS KOCK e.U., Vienna, Austria).

### 2.2. Cell Culture

The human colorectal carcinoma cell line HCT116 was kindly provided by Dr. Vogelstein from Johns Hopkins University, Baltimore. The pancreatic adenocarcinoma cell line Capan-1 was purchased from the American Type Culture Collection (ATCC, Manassas, VA, USA). HCT116 cells were cultured in McCoy’s 5A Modified Media (Sigma-Aldrich, St. Louis, MO, USA) supplemented with 10% heat-inactivated fetal calf serum (FCS, PAA, Linz, Austria) and 2 mM glutamine (Sigma-Aldrich, St. Louis, MO, USA). Capan-1 cells were grown in RPMI-1640 (Sigma-Aldrich, St. Louis, MO, USA) supplemented with 10% FCS. BOLD-100-resistant HCT116 and Capan-1 sublines were generated over several months by regular exposure to increasing concentrations of BOLD-100, followed by a drug-free recovery phase. Finally, BOLD-100-resistant HCT116 (HCTR) and Capan-1 (CapanR) cells were selected with 200 µM BOLD-100 in two-week-intervals. Resistance was maintained for >1 month without selection pressure. All cell cultures were grown in 5% CO_2_ at 37 °C and regularly checked for *Mycoplasma* contamination.

### 2.3. Cell Viability Assay

HCT116 and HCTR cells were seeded at a density of 3.5 × 10^3^ cells/100 µL and Capan-1 and CapanR cells were seeded at a density of 6 × 10^3^ cells/100 µM in 96-well microtiter plates and allowed to adhere overnight. Cells were exposed to BOLD-100, 2-DG, CQ, or their combination in the indicated concentrations for 72 h. Cell viability was determined using the 3-(4,5-dimethylthiazol-2-yl)-2,5-diphenyltetrazolium bromide (MTT) assay (EZ4U, Biomedica, Vienna, Austria), following the manufacturer’s recommendations. The colorimetric signal was measured spectrophotometrically at 450 nm and a reference wavelength of 620 nm, using the Asys Expert Plus micro plate reader (v1.4, Asys Hitech GmbH Hitech, Eugendorf, Austria). Data were analyzed using the GraphPad Prism software (La Jolla, CA, USA). IC_50_ values, indicating the drug concentration resulting in a 50% reduction of cell viability as compared to untreated control cells, were calculated from full dose-response curves by non-linear regression curve fitting (sigmoidal dose-response with a variable slope). For the evaluation of the synergistic effects of drug combinations, the CalcuSyn software (Biosoft, Ferguson, MO, USA) was used [39]. The respective effects are expressed as combination indices (CI). CI values indicate drug activity as follows: CI < 0.9, synergism; CI = 0.9–1.2, additive effects; or CI > 1.2, antagonism.

### 2.4. Clonogenicity (Clone Formation) Assay

To determine the clone formation capacity, HCT116 and HCTR cells were seeded at a density of 2000 cells/mL in 250 µL of medium in 24-well microtiter plates and allowed to adhere overnight. The cells were treated with the indicated concentrations of the BOLD-100, 2-DG, or their combination for 7 days. Cells were washed with ice-cold phosphate-buffered saline (PBS, pH 7.4 Sigma-Aldrich Inc., St. Louis, MO, USA), fixed with acetone and methanol in a ratio of 1:1 for 30 min., washed twice, and stained with 0.01% (*w*/*v*) crystal violet dissolved in ethanol for 5 min. Next, the plates were washed with ddH_2_O, dried, and the fluorescent signal (610/30 nm BP emission filter, 633 laser) of the stained cell clones was scanned with a Typhoon TRIO Variable Mode Manager Imager System (Amersham Bioscienes, UK) using the Typhoon Scanner Control software (V5.0). From the derived images, pixel intensities per well were quantified by Image J 1.50i (NIH, Bethesda, MD, USA).

### 2.5. Cellular Ruthenium Uptake Measurements

HCT116 and HCTR or Capan-1 and CapanR cells were seeded at a density of 1 × 10^5^ cells/mL in 1 mL of respective cell culture medium in 6-well plates in a total volume of 2 mL and left to recover overnight. Cells were treated in triplicates with 100 µM BOLD-100 for 24 h. BOLD-100 was additionally added to cell-free blank wells to determine the absorption of the compound to the plastic well. Sample processing, measurement of cellular ruthenium content, and data evaluation were performed as described previously [40]. In short, ^102^Ru standards were derived from Labkings (Hilversum, The Netherlands). Total ruthenium content was determined using a quadrupole-based inductively coupled plasmon mass spectrometry (ICP-MS) instrument Agilent 7800 (Agilent Technologies, Tokyo, Japan) equipped with the Agilent SPS 4 autosampler (Agilent Technologies, Tokyo, Japan) and a MicroMist nebulizer at a sample uptake rate of approximately 0.2 mL min^−1^. Argon was used as the plasma gas (15 L min^−1^) and carrier gas (1.06 L min^−1^). The integration time was set to 0.3 s with 10 replicates and 100 sweeps/replicate. ^115^In served as the internal standard for ruthenium. The Agilent MassHunter software package (Workstation Software, Version C.01.04, 2018) was used for data evaluation.

### 2.6. Total-RNA Isolation and Whole Genome Gene Expression Array

Whole genome gene expression arrays were performed as described previously [41]. Briefly, mRNA was isolated with the RNeasy Mini Kit (QIAGEN GmbH, Hilden, Germany) from HCT116, HCTR, Capan-1, and CapanR cells that were either left untreated or treated with 100 µM BOLD-100 for 6 h. Whole genome gene expression analyses were performed on 4 × 44 K oligonucleotide microarrays (G4845A, Agilent Technologies, Santa Clara, CA, USA) according to manufacturer’s recommendations. Feature extraction was carried out using the Feature Extraction software (version 11.5.1.1). Differentially expressed genes in BOLD-100-treated vs. untreated cells were analyzed using the GeneSpring software (version 13.0). Gene Set Enrichment Analysis (GSEA) on the loess (within arrays) and quantile (between arrays) normalized gene expression matrix was conducted on the C2 dataset of the molecular signature database (version 7.4) using the GSEA software (version 4.0.3) with default parameters and the log2FC as ranking metric [42,43].

### 2.7. Metabolomics

HCT116 and HCTR cells were seeded at a density of 0.25 × 10^6^ cells/mL in 1 mL medium in 12-well plates and left to recover overnight. The medium was replaced and the cells were treated with 100 µM BOLD-100 or the corresponding concentration of DMSO for 24 h. The cells were washed 3 times with PBS (pH 7.4), quenched with liquid nitrogen, and stored at −80 °C until further processing. The metabolomics experiments were carried out as described before [31]. In short, liquid chromatography high-resolution mass spectrometry (LC-MS) measurement with a Thermo Scientific Q Exactive HF quadrupole-Orbitrap mass spectrometer was utilized in full mass scan mode (both positive and negative ionization-mode) at a resolution of 120,000. External calibration with fully labelled ^13^C internal standards ISOtopic solutions (Vienna, Austria) was used for quantification. The obtained absolute metabolite amounts (pmol) were normalized to the total protein content in their respective well (µg) with the Thermo BCA kit, according to manufacturer’s instructions.

### 2.8. Gene-Metabolite Network Analysis

A pathway-based compound-gene network was constructed with the Cytoscape [44] plugin MetScape [45,46] based on annotated data from the Kyoto Encyclopedia of Genes and Genomes (KEGG) database [47,48,49] and significantly regulated compounds and genes (raw *p*-value < 0.05) were mapped on it. The metabolomics data were provided with raw *p*-values and fold-changes, based on pmol absolute amounts normalized to µg of protein content in the respective sample. While 5-methyluridine, *N*-acetyl-serine, methionine sulfone, and N4-acetylcytidine were not retrieved from KEGG, otherwise 46 unique compounds could be mapped on the network. The respective KEGG pathways were extracted as subnetworks for further analysis.

### 2.9. Western Blot Analyses

Groups of 1–2 × 10^6^ cells were seeded in 1 mL medium in 6-well plates in a total volume of 2 mL and allowed to adhere overnight. The cells were treated for 24 h with 100 µM BOLD-100, 1 or 10 mM 2-DG, 50 µM CQ, or their combination. Two solvent controls, medium without or with DMSO, were included. Cells were harvested by scratching into the supernatant, centrifugation, and washing with ice-cold PBS (pH 7.4). Sample collection, protein separation, and Western blotting were performed as described in detail previously [50]. The protein concentration was determined using the Micro BCA™ Protein Assay Kit (Thermo Fisher Scientific, Waltham, MA, USA). Sodium dodecyl sulfate polyacrylamide gel electrophoresis (SDS-PAGE) was performed to separate protein extracts. Subsequently, the proteins were transferred onto polyvinylidene difluoride membranes (PVDF, Thermo Fisher Scientific, Waltham, MA, USA). The primary antibodies Anti-GRP78 (C50B12) (#3177, dilution 1:1000), p-eIF2A (Ser51) (#9721S, 1:1000), eIF2A (L57A5) (#2103, 1:1000), HSP70 (#4872, 1:1000), PARP (46D11) (#9532, 1:1000), beclin-1 (#3495, 1:1000), p62/SQSTM1 (#8025, 1:500), and LC3 B I/II (#12741, 1:1000) were purchased from Cell Signaling Technology (Danvers, MA, USA). To obtain the horseradish-peroxidase-linked secondary antibodies, anti-Mouse IgG (Fc specific) antibody (A0168) was purchased from Merck KGaA (Darmstadt, Germany), and anti-rabbit IgG antibody (7074S) was purchased from Cell Signaling Technology (Danvers, MA, USA). Anti-ß-actin (AC-15) (A5441, 1:2000) was obtained from Sigma-Aldrich (St. Louis, MO, USA).

### 2.10. Flow Cytometry Analysis of Cellular 2-NBDG Uptake

HCT116 and HCTR cells were seeded at a density of 1 × 10^5^ cells/mL and Capan-1 and CapanR cells at a density of 2 × 10^5^ cells/mL in 1 mL medium 12-well plates and allowed to adhere overnight. The next day, the cells were washed with pre-warmed PBS (pH 7.4) and the medium was replaced with glucose-free RPMI-1640 medium containing L-glutamine (Sigma-Aldrich, St. Louis, MO, USA) supplemented with 10% FCS. The cells were treated with 100 µM BOLD-100 or the corresponding amount of DMSO (control) for 24 h. The medium with or without drug or solvent was replaced by serum-free medium containing 25 µM fluorescent d-glucose analog 2-[*N*-(7-nitrobenz-2-oxa-1,3-diazol-4-yl) amino]-2-deoxy-d-glucose (2-NBDG) (ab146200, Abcam, Cambridge, UK) and incubated in the dark for 1 h at 37 °C with 5% CO_2_ before flow cytometry analysis. 2-NBDG incubation was stopped by the removal of the incubation medium and careful washing of the cell layer with cold PBS (pH 7.4). The cells were detached by trypsinization, washed two times, and resuspended with FACS-PBS (7.81 mM Na_2_HPO_4_ × 2H_2_O, 1.47 mM KH_2_PO_4_, 2.68 mM KCl, and 0.137 M NaCl) and transferred into FACS tubes. The fluorescence intensity was measured on a LSRFortessa flow cytometer (BD Biosciences, East Rutherford, NJ, USA) with the FITC (530/30 nm) bandpass emission filter. The data were analyzed using the Flowing Software (University of Turku, Finland) and are depicted as fluorescence intensities (arbitrary units, a.u.).

### 2.11. Cell Cycle Analysis

For analysis of the cell cycle state, HCT116 and HCTR cells were seeded in triplicates at a density of 1.5 × 10^5^ cells/mL in 1 mL medium in 12-well plates and allowed to adhere overnight. Cells were treated with DMSO corresponding to BOLD-100 treatment, 100 µM BOLD-100, 10 mM 2-DG, or their combination for 24 h. Cells were trypsinized, washed with PBS (pH 7.4), and the resulting pellet was resuspended in 0.9% NaCl and fixed in ice-cold 70% ethanol by slow, drop-wise addition. Ethanol-fixed cells were stored at −20 °C until further use. Cells were centrifuged, the pellet was resuspended in FACS-PBS (7.81 mM Na_2_HPO_4_ × 2H_2_O, 1.47 mM KH_2_PO_4_, 2.68 mM KCl, and 0.137 M NaCl) containing RNAse (50 mg/mL, heat-inactivated) and 1 mg/mL propidium iodide (PI) (Sigma-Aldrich, St. Louis, MO, USA), and incubated for 30 min at 37 °C in the dark. The cells were measured by flow cytometry with a LSRFortessa flow cytometer (BD Biosciences, East Rutherford, NJ, USA) with the 610/20 nm bandpass emission filter. The percentages of 50.000 cells in G0/G1, S, G2/M, and <G1 or >G2 phase of the cell cycle were calculated by FlowJo software (v10.06) using the Watson Pragmatic algorithm.

### 2.12. Cell Proliferation Assay

HCT116 and HCTR cells were seeded in triplicates at a density of 3 × 10^3^ cells/100 µL medium in 96-well plates and allowed to adhere overnight. The cells were exposed to DMSO, BOLD-100, 2-DG, or their combination in the indicated concentrations for 24 h. 5-ethynyl-2′-deoxyuridine (EdU) incorporation into de novo synthesized DNA was determined using the Click-iTTM EdU proliferation assay for microplates (C10499, Invitrogen, Waltham, MA, USA) according to manufacturer’s recommendations. The fluorescence was measured on a Tecan infinite 200Pro (Zurich, Switzerland) micro plate reader at 520/585 nm. The data were analyzed using the GraphPad Prism software (La Jolla, CA, USA). Antiproliferative IC_50_ values indicated the drug concentration resulting in a 50% reduction of EdU incorporation, reflecting the proliferation rate as compared to untreated control cells.

### 2.13. Hoechst 33258/PI Staining

HCT116 and HCTR cells were seeded in triplicates at a density of 4 × 10^4^ cells/mL in 1 mL medium in 24-well plates and allowed to adhere overnight. The cells were exposed to DMSO, BOLD-100, 2-DG, or their combination in the indicated concentrations for 72 h. After the drug incubation period, the cells were stained with 2 µL/mL of a 1:1 mixture of Hoechst 33258 (1 mg/mL in PBS (pH 7.4); termed Hoechst in this manuscript) and PI (2.5 mg/mL in PBS pH 7.4) for 1 h at 37 °C. Microscopic images were taken using a Nikon Eclipse Ti inverted microscope (Visitron Systems, Puchheim, Germany) with the DAPI (Hoechst) and Cy3 (PI) filters. Light and evenly distributed blue fluorescence on PI-negative cells indicated living cells. A bright blue fluorescent signal of condensed chromatin on PI-negative cells indicated mitotic nuclei together with mitosis-characteristic features. Bright blue fluorescence of condensed nuclei associated with the formation of apoptotic bodies in the absence of a PI signal was considered indicative for early apoptotic cells. PI-positive cells were considered to be dead cells with late apoptotic nuclei.

### 2.14. High-Resolution Live-Cell Spinning-Disc Imaging of Lysotracker Red and Hoechst 33258-Stained Cells

HCT116 and HCTR cells were seeded at a density of 0.5 × 10^5^ cells/mL in 200 µL medium in 8-well ibiTreat 1.5 polymer µ-slides (80826, ibidi GmbH, Gräfelfing, Germany) and allowed to adhere overnight. The cells were exposed to DMSO, BOLD-100, 2-DG, or their combination in the indicated concentrations for 24 or 72 h. After the drug incubation period, the cells were stained with 0.5 µM Lysotracker Red (L7528, Life Technologies, CA, USA) and Hoechst 33258 (blue) (1 mg/mL in PBS pH 7.4, Sigma-Aldrich). High-resolution live-cell spinning-disc images were taken using an IX83 P2ZF microscope (Olympus Austria GmbH, Vienna, Austria) with a UPLXAPO O 60 oil-immersion objective at 192× magnification. Image acquisition was performed using the OLMYPUS cellSens Dimension 3.1 imaging software (OLYMPUS CORPORATION, Tokio, Japan).

### 2.15. High-Resolution Live-Cell Spinning-Disc Imaging of LC3B II Foci

HCT116 and HCTR cells were seeded at a density of 0.5 × 10^5^ cells/mL in 200 µL medium in 8-well ibiTreat 1.5 polymer µ-slides (80826, ibidi GmbH, Gräfelfing, Germany) and allowed to adhere overnight. The cells were exposed to DMSO, BOLD-100, 2-DG, or their combination in the indicated concentrations for 24 h. The cells were fixed with 4% paraformaldehyde (PFA)/PBS (pH 7.4) for 30 min. and permeabilized in PBS/0.5% *v*/*v* Triton X-100 (pH 7.4) for 20 min at room temperature. The cells were blocked in PBS (pH 7.4) containing 20% *w*/*v* BSA for 20 min and stained with primary anti-LC3B rabbit monoclonal antibody (1:400, #12741 from Cell Signaling Technology, Danvers, MA, USA) in PBS (pH 7.4) containing 2% *w*/*v* BSA at 4 °C overnight. The cells were washed in PBS (pH 7.4), incubated with Alexa Fluor 488-labeled goat anti-rabbit IgG, (1:500; Invitrogen, Thermo Fisher, USA) for 1 h at room temperature in the dark, and nuclei were counterstained with Hoechst 33258 (blue) (1 mg/mL in PBS pH 7.4, Sigma-Aldrich). After a final washing step, high-resolution live-cell spinning-disc images were taken, as described above.

### 2.16. Flow Cytometry Analysis of Lysotracker-Red-Stained Cells

HCT116 and HCTR cells were seeded at a density of 0.5 × 10^5^ cells/mL in 1 mL medium in 12-well plates and allowed to adhere overnight. The cells were treated with DMSO, BOLD-100, 2-DG, or their combination in the indicated concentrations for 72 h. The cells were stained with 0.5 µM Lysotracker Red (L7528, Life Technologies, CA, USA) for 30 min at 37 °C. The cells were prepared and measured, as described above for flow cytometry analysis, and measured with the 610/20 nm bandpass emission filter. The data were analyzed using the Flowing Software (University of Turku, Finland) and are depicted as fluorescence intensities.

### 2.17. Mass Spectrometry Analysis

Stock solutions of 20 mM d-glucose in H_2_O and 20 mM BOLD-100 in DMSO were prepared and diluted with H_2_O for coincubation of 100 µM BOLD-100 and 400 µM d-glucose. The reaction mixture was incubated at 37 °C in the dark under constant shaking. Reaction aliquots were taken after 30 min and 2, 4, and 24 h and immediately snap frozen with liquid nitrogen and stored at −20 °C. The data were acquired on a maXis QTOF mass spectrometer (Bruker Daltonics, Bremen, Germany) by the direct infusion of analytes diluted to 5 µM with an aqueous solution (LC-MS grade water, Fluca). The infusion rate was 3 µL min^−1^. The instrument parameters were as follows: 4.5 kV capillary voltage, 500 V end plate offset, 3 bar nebulizer, 5 L min^−1^ dry gas, 180 °C dry temperature, and *m*/*z* 50–1200 scan range. The mass spectra were recorded in the negative ion mode over 0.4 min and averaged.

### 2.18. Lipidomics

HCT116 and HCTR cells were seeded at a density of 0.5 × 10^6^ cells/mL in 1 mL medium in 6-well plates in a total volume of 2 mL and allowed to adhere overnight. Cells were treated for 24 h with 100 µM BOLD-100 or DMSO as solvent control (*n* = 3 per condition). The medium was removed, and the samples were washed three times with PBS (pH 7.4). All samples were quenched with liquid nitrogen and stored at −80 °C prior to extraction. ^13^C-labeled LILY lipids were added for internal standardization [51]. A two-phase extraction with MTBE was used to collect (1) the upper lipid-containing phase and (2) the protein pellet for protein normalization [52]. The supernatant was dried and dissolved in 100 µL of isopropanol prior to LC-MS analysis. Lipidomics measurements were performed with reversed-phase (RP) chromatography (C18 column) and a 27 min isopropanol gradient coupled to a high-resolution mass spectrometer (HRMS) (Orbitrap ID-X, Thermo Fisher Scientific, Waltham, MA, USA). A resolution of 120,000 was used for MS1 followed by a data-dependent MS2 acquisition and deep scans (Acquire X) at a resolution of 30,000 in positive and negative ionization mode. Targeted and non-targeted data evaluation was based on LipidSearch 5.0 (Lipid Search 5.0.63, Thermo Fisher Scientific, Waltham, MA, USA), Skyline (21.1.0.278, https://skyline.ms/project/home/begin.view, accessed on 2 January 2022), and R Studio (R version 4.1.2, Boston, MA, USA).

### 2.19. Statistical Analysis

Data were analyzed using GraphPad Prism software (version 8.0.1, La Jolla, CA, USA). If not stated otherwise in the figure legends, one out of at least three independent experiments in triplicates is displayed. Each data point represents the mean ± SD of triplicate values. The statistical evaluation of significance was performed using an unpaired Student’s *t*-test, as well as a one- or two-way analysis of variance (ANOVA) with Bonferroni’s or Tukey’s multiple comparisons post-tests. *p*-values below 0.05 were considered statistically significant: * *p* < 0.05, ** *p* < 0.01, *** *p* < 0.001, **** *p* < 0.0001.

## 3. Results

### 3.1. Impact of BOLD-100 Exposure and Acquired Resistance on Cellular Carbohydrate Metabolism

Since BOLD-100 is currently being investigated clinically (NCT04421820) and the development of chemotherapy resistance remains a major hurdle in cancer treatment [53,54], this study aimed to characterize the mechanisms underlying acquired BOLD-100 resistance. Therefore, the intrinsically BOLD-100-sensitive colon cancer HCT116 cell line and pancreatic Capan-1 cancer cell line [33] were selected for acquired drug resistance via continuous exposure to increasing concentrations of BOLD-100 up to 200 µM. Representative phase contrast images depict the changes of cell morphology induced by treatment with 100 µM BOLD-100 in acquired -resistant (HCTR or CapanR) as compared to parental cells (Appendix A). While the pronounced appearance of shirked and fragmented cells, indicative for apoptosis induction, was recorded in both parental cell lines, these effects were distinctly reduced or even absent in the resistant sublines. This corresponded well with a significantly decreased sensitivity of the BOLD-100-resistant sublines in the MTT assay, reflected by increased IC_50_ values (Figure 1a). Overall, the selection resulted in 2.2-fold increased BOLD-100 resistance in HCTR (IC_50_ of HCTR vs. HCT116 cells) and a 4.1-fold increased resistance in CapanR cells. Whole genome gene expression analyses of drug transporter expression in BOLD-100-resistant as compared to parental cells (Appendix A) revealed a slight but significant increase in ABCB1, a decrease in ABCC1, and no change in the case of ABCG2 in HCTR vs. HCT116 cells. In CapanR as compared to Capan-1 cells, no significant drug transporter regulation was detected. Accordingly, the measurement of intracellular ruthenium accumulation by ICP-MS demonstrated that acquired BOLD-100 resistance was not based on a decreased drug uptake (Figure 1b). The data of whole genome gene expression analyses of acquired BOLD-100-resistant vs. respective parental HCT116 or Capan-1 cells were processed by GSEA. Of the top 20 gene sets enriched in the acquired BOLD-100-resistant vs. sensitive HCT116 cells, seven gene sets were associated with carbohydrate metabolism. From the KEGG database, “GALACTOSE_METABOLISM” (nominal *p*-value < E^−7^; FDR 0.010) and “FRUCTOSE_AND_MANNOSE_METABOLISM” (nominal *p*-value < E^−7^; FDR 0.084) were identified as the third and fourth most significantly enriched gene sets, respectively, in HCTR vs. HCT116 cells (Figure 1c). Exemplarily, both heat maps show an upregulated expression of the mRNA of hexokinase genes, translating into enzymes catalyzing the first and rate-limiting step in glycolysis. Accordingly, upon BOLD-100 treatment in HCT116 cells “GLYCOSAMINOGLYCAN_METABOLISM” (nominal *p*-value < E^−7^; FDR 0.003) was identified as the fourth most down-regulated gene set in the REACTOME database and “O_GLYCAN_BIOSYNTHESIS” (nominal *p*-value < E^−7^; FDR 0.003) as the seventh most down regulated in the KEGG database (Appendix A). The GSEA of the acquired CapanR vs. Capan-1 cells identified “KEGG_PYRUVATE_METABOLISM” (nominal *p*-value 0.147; FDR 0.596) as the eighth and “KEGG_N_GLYCAN_BIOSYNTHESIS” (nominal *p*-value 0.193; FDR 0.703) as the tenth most significantly enriched gene set (Figure 1d). Screening of the mRNA data identified a higher expression of the solute carrier family 2 and facilitated glucose transporter SLC2A4 and SLC2A9 genes in HCTR, as well as a higher expression of SLC2A9 and SLC2A12 genes in CapanR as compared to their respective parental cell lines (Figure 1e). Taken together, these data show the transcriptional changes associated with cellular carbohydrate metabolism in response to acute treatment with and acquired resistance against BOLD-100. This suggests a possible role of carbohydrate metabolism as a defense/resistance mechanism against the ruthenium compound.

### 3.2. Enhanced Glucose Uptake and Glycolytic Activity in Acquired BOLD-100-Resistant Cells Translate into a Metabolic Vulnerability Targetable by 2-DG

Consequently, we hypothesized that the changes in glycolysis-associated gene expression should translate into an altered metabolic phenotype. Hence, BOLD-100-sensitive and –resistant cells were exposed to the fluorescent glucose analog 2-NBDG to measure and quantify glucose uptake with or without treatment with BOLD-100. Indeed, 2-NBDG uptake was significantly increased in HCTR cells as compared to parental cells (Figure 2a). In HCT116 cells, BOLD-100 treatment significantly increased intracellular 2-NBDG levels, while it was decreased in HCTR cells (Figure 2b). These results indicate both an enhanced glucose uptake as a consequence of acquired BOLD-100 resistance and a different response state to BOLD-100 treatment concerning glucose uptake as a consequence of resistance development. Concerning the pancreatic cancer cell models, CapanR cells showed, again, a significantly higher uptake of 2-NBDG as compared to Capan-1 cells and a distinct increase in glucose uptake induced by BOLD-100 treatment in the sensitive parental but not in the resistant subline (Appendix A). Based on these results, extracellular formation of a potential glucose-BOLD-100 adduct with different transport characteristics than free glucose might be postulated. However, MS analysis demonstrated that under cell-free conditions BOLD-100 did not form any adduct with glucose (Appendix A), arguing against this hypothesis. For further in-depth investigation, gene-metabolite networks were established by a combined computational analysis of the whole genome gene expression and metabolomics data from HCTR as compared to HCT116 cells. Based on these network data, a clear-cut activation of glycolysis with acquired BOLD-100 resistance was suggested (Figure 2c). Besides the above-mentioned HK genes (compare Figure 1c), phosphofructokinase genes PFKP and PFKFB4, coding for the most important rate-limiting enzymes in glycolysis, were significantly upregulated together with clearly enhanced levels of ß-D-fructose-1,6-bisphosphate, an activator of pyruvate kinase [55]. Consequently, the level of the downstream glycolysis product pyruvate was massively increased in HCTR vs. HCT116 cells, together with a decrease in cellular lactate contents (Figure 2d). Treatment with BOLD-100 distinctly reduced pyruvate and lactate levels in HCT116 cells, while this effect was clearly reduced for pyruvate and completely lost in case of lactate in HCTR cells. The above-described stimulation of glucose uptake upon BOLD-100 treatment in HCT116 cells which, however, was paralleled by decreased pyruvate and lactate levels, indicates an interference of BOLD-100 with glycolysis. This hypothesis was supported by downregulated HK2 mRNA expression upon BOLD-100 treatment in HCT116 and Capan-1 cells (Appendix A).

Together, the identified metabolic perturbations in cells with acquired BOLD-100 resistance led us to the hypothesis that the resistance phenotype might be characterized by novel metabolic vulnerabilities. Hence, glycolysis was inhibited pharmacologically with 2-DG and the effects on cell viability were assessed. Indeed, HCTR (Figure 3a) and CapanR (Appendix A) cells were distinctly hypersensitive to 2-DG, while inhibition of glutaminolysis with BPTES [56] did not lead to any differences (Figure 3b). This suggests that HCTR cell survival depends on a functional glucose metabolization but not on glutamate utilization. These data clearly show that increased cellular glycolysis plays a key role in acquired resistance against BOLD-100 in HCT116 as well as Capan-1 cells.

### 3.3. Glucose Deprivation by 2-DG Presents a Novel Strategy to Boost BOLD-100 Anticancer Activity and Overcome Acquired Resistance

Accordingly, we hypothesized that the inhibition of glycolysis might present a potential strategy to revert acquired BOLD-100 resistance. In this regard, cells were concomitantly treated with 2-DG and BOLD-100. Morphologically, BOLD-100-treated HCT116 cells appeared to be massively fragmented, indicative of apoptosis induction, while this effect was strongly reduced in HCTR cells (Appendix A). Conversely, 2-DG treatment for only 24 h reduced the cell layer density, especially of HCTR cells and to a lesser extent of HCT116 cells. The combination of both compounds led to cell detachment, especially of the parental cells, and further reduced cell density in both sublines. Concerning the cell viability assay results, low concentrations of 2-DG antagonized low-dose cytotoxicity of BOLD-100 in HCT116 cells. Higher concentrations enhanced the cytotoxic activity (Figure 3c). In HCTR cells, already low doses of 2-DG amplified the anticancer activity of BOLD-100. This was verified by the calculation of CI values according to the method published by Chou and Talalay [57]. CI values depicted additive to synergistic activity of the two compounds in both cell models (Figure 3d). Additionally, the clone formation capacity under long-term treatment with 2-DG/BOLD-100 was tested (Figure 3e). As expected, BOLD-100 alone was highly effective in HCT116 cells, while HCTR cells were more resistant to the ruthenium drug. In line with the cell viability assays, 2-DG was more active against clone formation in HCTR cells than in HCT116 cells. The combination of 2-DG and BOLD-100 reduced the clone formation ability of HCTR cells by 31% and 53% as compared to treatment with 2-DG or BOLD-100 alone, respectively. In HCT116 cells, the effects against clone formation were mainly attributed to the already high activity of BOLD-100. Accordingly, an additive to synergistic cytotoxic activity of 2-DG with BOLD-100 was also observed in Capan-1 and CapanR cells (Appendix A).

Consequently, we wondered which mechanisms underlie the hypersensitivity of HCTR cells against 2-DG and the synergistic activity of the combination with BOLD-100 in both cell lines. Previously described cellular responses to 2-DG include cell cycle arrest or reduction of the proliferation rate, both with or without apoptosis induction [16], as well as enhanced autophagy [58]. Thus, we tested the impact of BOLD-100, 2-DG, or their combination on cell cycle distribution (Appendix A) and cellular proliferative potential via EdU incorporation during DNA synthesis (Appendix A). Generally, HCTR cells contained a higher percentage of cells in G2/M phase. Fitting to observations from cell survival assays (compare Figure 3a,c,e), BOLD-100 decreased S and increased G0/G1 phase fractions. In line with this, the proliferation rate of HCT116 cells was dose-dependently reduced by BOLD-100 treatment, in agreement with previous observations [33]. These effects were less pronounced in HCTR cells with a proliferation rate reduction by a factor of 2 (antiproliferative IC_50_ in HCTR: 90.74 µM vs. HCT116 cells: 47.95 µM, Appendix A). In both cell models, 2-DG increased G0/G1 and reduced S phase, fitting to DNA synthesis rate reduction (antiproliferative IC_50_ in HCTR: 6.20 µM vs. HCT116 cells: 5.76 µM). Unexpectedly, in HCT116 cells the combination treatment counteracted the single drug-induced G0/G1 arrest and increased the G2/M phase fraction. Conversely, in HCTR cells the combination treatment induced a G0/G1 phase increase and reduced the S fraction. While in both cell models the combination treatment synergistically and dose-dependently reduced proliferation, no differences in the single drug activity of 2-DG were found. This rules out an altered impact on DNA synthesis as a factor underlying HCTR cell hypersensitivity to 2-DG.

Next, we performed apoptosis assays via Hoechst/PI co-staining of HCT116 and HCTR cells after 72 h of treatment with BOLD-100, 2-DG, or their combination (Figure 4a, quantification of early or late apoptotic cells in Figure 4b). In the solvent controls of HCT116 and HCTR, the vast majority of cells with condensed chromatin (intensified blue Hoechst staining) were PI-negative and exhibited a mitotic morphology. Conversely, only a few cells contained compacted apoptotic chromatin and were generally PI-positive (red signals; late-apoptotic and dead cells). Upon treatment with BOLD-100, HCT116 cell viability was massively reduced and multiple cells displayed compacted apoptotic nuclei with subpopulations being PI-negative (early apoptotic) or -positive. In HCTR cells, these effects were also observed but to a much lesser extent. 2-DG treatment strongly decreased HCTR cell growth as compared to DMSO-treated controls, however with only minor apoptosis induction. It is noteworthy that it dramatically affected cell morphology. Surviving HCTR cells displayed cell swelling, multinucleation, and massive cytoplasmic vacuolization (Appendix A). In contrast, HCT116 cell morphology under 2-DG treatment appeared widely unchanged with only minor cytoplasmic vacuole formation. Concomitant 2-DG and BOLD-100 treatment in both cell lines significantly induced cell death induction by apoptosis. While surviving HCT116 cells in the combination treatment were bigger with enlarged nuclei, surviving HCTR cells again exhibited distinct morphological changes and massive vacuolization (Appendix A).

### 3.4. HCT116 and HCTR Cells Differ in ER Stress Response to 2-DG and BOLD-100 Treatment

2-DG, based on glucose deprivation, is known to induce ER stress via the prevention of *N*-glycosylation of proteins, leading to improper folding and trafficking and, finally, induction of the UPR [19,59,60]. Likewise, BOLD-100 causes enhanced ER stress, at least in part by interfering with ER chaperone GRP78 expression [32,33,61]. Hence, we investigated the interplay of these two agents at the level of the UPR and ER stress signaling by detecting chaperone expression and phosphorylation of eIF2A. The latter process halts protein translation to repair improperly folded proteins and, hence, restore protein homeostasis [62]. Respective Western blot analyses from single and combination treatment samples are shown in Figure 4c. Two different solvent control groups were included: (1) the medium control representing the background of 2-DG treatment and (2) the DMSO-containing control for the BOLD-100 matrix. BOLD-100 enhanced the phosphorylation of eIF2A at serine 51 but reduced basal ER chaperone GRP78 expression in HCT116 cells. In accordance with the literature [61,63], these data prove uncoupling of ER stress induction and chaperone upregulation as a consequence of BOLD-100 treatment. The ER stress response was accompanied by a cleavage of the apoptosis marker PARP [64]. In contrast, the expression of GRP78 was unchanged by BOLD-100 treatment in HCTR cells, while the increase of eIF2A (Ser51) phosphorylation level was comparable to the parental cell line. Consistent with cell viability (compare Figure 1a), the resistant subline lacked BOLD-100-induced PARP cleavage. As expected [21], 2-DG treatment increased glucose-sensing GRP78 protein levels in the BOLD-100-sensitive as well as the BOLD-100-resistant cell model. While in HCT116 cells peIF2A levels were even reduced in response to 2-DG, in HCTR cells peIF2A levels distinctly increased. This indicates a hyperactivation of the glucose restriction-mediated ER stress response as a consequence of acquired BOLD-100 resistance. In HCTR cells, no enhanced cleavage of PARP was detected upon 2-DG treatment. This suggested that the hypersensitivity of HCTR cells against 2-DG (compare Figure 3a) was based on other mechanisms than apoptosis induction. Concerning the drug combination experiments, BOLD-100 treatment counteracted 2-DG-induced GRP78 upregulation in HCT116 cells. Additionally, 2-DG treatment reduced BOLD-100-induced eIF2A phosphorylation. In sharp contrast to the parental cells, combined BOLD-100 and 2-DG treatment further enhanced GRP78 expression with steady peIF2A levels in HCTR cells but without the cleavage of PARP. To further evaluate the cellular response to the tested cellular stressors, HSP70 protein expression was investigated (Figure 4c). HSP70, like GRP78, belongs to the family of stress-responsive HSP70 proteins and is implicated in the maintenance of cellular protein homeostasis in an ATP-dependent manner [65]. While in HCT116 cells HSP70 levels remained widely unchanged, even under drug exposure, 2-DG treatment efficiently reduced HSP70 levels in HCTR cells. Interestingly, the combination with BOLD-100 even potentiated this effect. The loss of HSP70 expression under 2-DG treatment specifically in HCTR cells might be related to the observed hypersensitivity to glucose deprivation (compare Figure 3a).

### 3.5. Cytoplasmic Lysosome HyperAccumulation Presents a Survival Mechanism against Glucose Deprivation by 2-DG

Since ER stress induction via phosphorylation of eIF2A at serine 51 is a central event in autophagy initiation [66], we hypothesized that the observed 2-DG-induced vesicle formation might be related to a deregulation of the autophagic machinery [58]. In general, autophagy is a ubiquitously executed catabolic process ensuring cellular homeostasis via the sequestration of damaged proteins and organelles into autophagosomes and their degradation upon fusion with lysosomes [67]. From the KEGG database, “Lysosome” (nominal *p*-value < E^−7^; FDR 0.01) was the fifth most enriched gene set in HCTR cells as compared to HCT116 cells (Figure 5a, right). Exemplarily, lysosome-associated membrane protein (LAMP) family mRNA expression was strongly increased in HCTR cells (heat map in Figure 5a, left). Consequently, the impact of time-dependent single and combination treatments on lysosomal dynamics was investigated by Lysotracker (red) staining via flow cytometry (Figure 5b, left) and high-resolution live-cell spinning-disc imaging (Figure 5c). In agreement with the GSEA results, an increased lysosome content of HCTR cells as compared to HCT116 cells was suggested by both methods. Survivor HCT116 cells of acute treatment with BOLD-100 (especially at 72 h exposure) increased the number of lysosomes, while in already lysosome-enriched HCTR cells Lysotracker accumulation remained widely unchanged, indicating a vital role of regulation of lysosome dynamics in the mode of action of BOLD-100. Surviving cells of glucose deprivation by 2-DG for 72 h strongly increased the lysosomal compartment in both cell lines as compared to the respective controls, with the highest lysosomal contents, by far, achieved in the HCTR cells. In the combination setting, BOLD-100 counteracted the 2-DG-induced increase of lysosomes in the case of HCT116 cells but not in HCTR cells. In addition to the altered lysosomal compartment, the massive differences in the impact on cell morphology and vacuolization (compare Figure 4a) were confirmed. Hence, survivors of 2-DG treatment induced a switch to a more mesenchymal, fibroblastoid appearance in HCT116 cells, while HCTR cells displayed a massive retraction of cell extensions and distinct cell clustering. Moreover, Lysotracker-positive lysosomes appeared to encircle as a layer the 2-DG-induced smaller Lysotracker-negative vacuoles, especially in HCTR cells (Figure 5c, white arrows).

### 3.6. 2-DG Reduces Cytoplasmic Lysosome Accumulation and Deregulates Autophagic Processing

Considering the massive induction of the lysosomal compartment observed after 72 h as potential survival mechanism against 2-DG, we treated cells for only 24 h with the aim to capture the cell population before cell death induction. Again, the impact of these short-time single and combination treatments on lysosomal dynamics was investigated by Lysotracker (red) staining evaluated by flow cytometry (Figure 5b, right) as compared to high-resolution live-cell spinning-disc imaging (Figure 5d). BOLD-100 increased the number of lysosomes in both cell models, with HCT116 cells already starting to detach from the surface (Figure 5d, inset; compare Figure 4). Supporting our hypothesis on the enrichment of the lysosomal compartment as a survival mechanism against 2-DG, glucose deprivation for 24 h decreased the number of lysosomes in HCT116 and HCTR cells. Morphologically, HCTR cells displayed the massive vacuolization already at this earlier time point (white arrows in Figure 5d), again, distinctly clustering with lysosomes. In the combination setting, BOLD-100 counteracted the 2-DG-induced decrease of lysosomes, especially in the resistant cell model. This antagonism was accompanied by a pronounced accumulation of larger lysosomes. Consequently, the regulation of the autophagic machinery in response to the investigated treatments was assessed on the marker protein level (Figure 6a). Beclin-1 binds to several co-factors forming core complexes that have a key role in the initiation of autophagosome formation [68]. p62/SQSTM1 is implicated in the formation of ubiquitinated protein aggregates that are degraded in the autophagic process [69,70]. LC3B II is the phosphatidylethanolamine (PE)-conjugated form of LC3B I that is incorporated in the membrane during autophagosome formation [71]. LC3B II foci formation, as a marker for the number of autophagosomes, was tested under respective treatment conditions and visualized using high-resolution live-cell spinning-disc imaging (Figure 6b). The ratio of LC3B II/I was calculated (Figure 6d) to monitor the autophagic flux. A reduction of both p62 and beclin-1 levels by BOLD-100 indicates either promotion of the autophagic flux or complete downregulation of the autophagic cascade in HCT116 cells (Figure 6a). Unexpectedly, BOLD-100 led to a distinct increase of both forms of LC3 B, however, with a reduced ratio of LC3B II/I. Together, these observations suggest the upregulation of autophagic flux by BOLD-100, however, paralleled by impaired execution of LC3B processing. Lipidomics analysis (Appendix A) revealed a trend towards a reduction of cellular PE levels by BOLD-100 treatment, delivering a potential mechanistic explanation for the defective LC3B processing in HCT116 cells. In HCTR cells, LC3B II remained widely unchanged and beclin-1 and p62 levels accumulated, the latter by a factor of 1.8 (black numbers in Figure 6a). Accordingly, lipidomics showed that PE levels were enriched in HCTR as compared to HCT116 cells. This suggests an adaptation of the autophagic machinery as a resistance mechanism against BOLD-100. To confirm the impact of BOLD-100 on autophagic flux, combination experiments under downstream blockade by 50 µM CQ, an inhibitor of lysosome-autophagosome fusion [72], were performed (Appendix A). As expected, CQ treatment alone blocked autophagy comparably in both cell models, as indicated by a p62 increase and an accumulation of LC3B II. BOLD-100 potentiated p62 accumulation in response to CQ in HCT116, supporting an enhancement of the autophagic flux. Again, LC3B delivered contradictory data with an increase at the level of unprocessed LC3B I but not LC3B II, in accordance with the supposed BOLD-100 impact on LC3B lipidation in HCT116 cells. In the case of HCTR cells, BOLD-100 in combination with CQ did not induce further p62 upregulation, while the impact on LC3B conversion, absent without CQ, was recovered in the combination setting. Glucose deprivation by 2-DG increased beclin-1 and p62 as well as LC3B levels in both models (Figure 6a,c and Appendix A) together with the formation of LC3B II foci (Figure 6b), all indicative of stimulation of the autophagic process [73]. These effects were more distinct in HCTR cells, especially concerning the LC3B II increase, e.g., by a factor 1.5 at 10 µM 2-DG. Accordingly, only in HCTR cells 2-DG dramatically increased the LC3B II/I ratio. The combination with CQ enhanced the effects of 2-DG alone regarding p62 accumulation and increased LC3B II in HCT116 cells (Appendix A). In HCTR cells, the combination with 2-DG did not further enhance CQ-induced LC3B II but, resembling the parental cell line under BOLD-100 treatment, increased LC3B I levels. The combination with BOLD-100 counteracted the 2-DG-induced accumulation of beclin-1 and p62, suggesting a more rapid processing of 2-DG-induced autophagy. Together this led to a massively induced presence of both LC3 forms in HCT116 cells, however, at a reduced LC3 B II/I ratio. Likewise, in HCTR cells, BOLD-100 combined with 2-DG increased both LC3B isoforms and decreased the LC3B II/I ratio. This suggests restored BOLD-100 autophagy inhibition at the level of LC3 conversion under glucose starvation. Concerning the hypersensitivity of HCTR cells against 2-DG, the combination with CQ delivered antagonizing effects on cell viability in HCTR cells but not in HCT116 cells (Figure 6e). This indicates that deregulated autophagy is responsible for the enhanced sensitivity of HCTR cells against glucose starvation, based on the accumulation of deglycosylated, misfolded proteins.

Together our data suggest that BOLD-100 interferes with the complex interplay between ER-stress response, lysosome dynamics, and autophagy execution. Acquired BOLD-100 resistance is accompanied by defective autophagy execution and hypersensitivity to 2-DG by disturbing lysosome-autophagosome fusion and hindering autophagic processing. Hence, the combination of both compounds represents an interesting strategy to synergistically enhance the activity of both therapeutic approaches.

## 4. Discussion

Glucose is an essential fuel for cellular survival and a factor driving malignant transformation [9,74]. Enhanced cellular glycolytic activity denotes poor prognosis in several solid tumor types and is often implicated in the development of intrinsic and acquired therapy resistance [5,7,8]. Hence, new strategies to specifically target malignant tissue and disable glycolytic activity are urgently needed. Insights into the metabolic alterations in response to ruthenium-based anticancer drugs and acquired resistance against this therapy remain sparse. Recently, we demonstrated that the metabolite abundances and metabolic flux alterations underlying acquired platinum- and ruthenium-based chemotherapy are severe and differ distinctly [31]. The dissection of such specific metabolic changes and their role in the resistance phenotype might open up approaches for resistance circumvention or even allow identification of novel therapeutic vulnerabilities.

In the present study, we have identified specific mechanisms associated with acquired BOLD-100 resistance in colon (HCT116) and pancreatic (Capan-1) cancer cells. In line with previous reports suggesting that BOLD-100 is not a drug transporter substrate [75,76], BOLD-100 resistance was not associated with altered intracellular drug accumulation. These data implicate that the acquisition of BOLD-100 resistance, at least in the two cell models tested, is not primarily based on altered influx or efflux mechanisms. Accordingly, whole genome gene expression analyses revealed no dramatic upregulation in drug transporter gene expression in response to BOLD-100 selection. A significant enrichment of carbohydrate-metabolism-associated genes was identified consistently in both BOLD-100-selected cell models. These mRNA expression changes corresponded with a distinctly enforced glycolytic activity, associated with an enhanced uptake of 2-NBDG, a fluorescent glucose mimic. 2-NBDG, as a substrate for glucose transporters, is used for direct glucose uptake measurement [77]. BOLD-100 treatment in HCT116 cells distinctly stimulated 2-NBDG uptake but, unexpectedly, markedly reduced intracellular pyruvate and lactate levels. As BOLD-100 was not interacting directly with glucose under cell-free conditions, these data strongly suggest a direct interference of the ruthenium compound with glycolysis. The underlying molecular mechanisms are enigmatic so far. Future experiments will address the question of whether the above mentioned rapid downregulation of HK2 mRNA expression in HCT116 and Capan-1 cells upon BOLD-100 treatment is one key factor underlying the observed glycolysis interference. Recent findings in T cells question if measurement of cellular 2-NBDG accumulation is, indeed, sufficient to assess cellular glucose uptake [78]. Hence, we decided to confirm the enhanced glucose metabolization in BOLD-100 resistance by combining metabolomics with the genome-wide expression data in a gene-metabolite network. This in silico approach again strongly supported an enhanced glycolytic activity in HCTR as compared to the sensitive parental cells, thus confirming the 2-NBDG uptake data.

Several studies have demonstrated an important role of altered glucose metabolism in the development of (chemo)therapy resistance [5]. Metabolic reprogramming in different platinum-, molecular targeted therapy-, or 5-fluorouracil-resistant tumor models translated into a metabolic vulnerability targetable by glycolysis inhibitors, such as 2-DG [11,30]. Accordingly, glycolysis interference by 2-DG was tested in our acquired BOLD-100-resistant cell models and, indeed, the enhanced glycolytic activity conferred hypersensitivity to 2-DG. The modes of action of 2-DG are manifold and vary distinctly in different cell models. In vitro responses to 2-DG include, amongst others, cell cycle arrest, reduction of the proliferation rate, both with or without apoptosis induction [16], regulation of epithelial-to-mesenchymal transition [30], energy stress, and growth inhibition due to ER stress [14], as well as regulation of autophagy [58]. Consequently, the question arose which factors drive HCTR cell hypersensitivity against glucose depletion by 2-DG. Together our data show that 2-DG treatment in sensitive and acquired BOLD-100-resistant HCT116 cells perturbs cell cycle progression and has antiproliferative activity at a widely comparable level, excluding general uptake/accumulation differences, as major contributing factors. One potential explanation lies in the above-mentioned rapid loss of HK2 mRNA expression under acute BOLD-100 treatment and its distinct overexpression after resistance development in HCTR cells as compared to parental cells. This suggests the induction of a stable glucose dependency by the ruthenium compound, which is present even in the absence of BOLD-100. Consequently, 2-DG-induced glucose deficiency led to distinctly higher cell death rates in HCTR as compared to parental cells. This cell-death-inducing effect was further enhanced upon the combination of 2-DG with BOLD-100 in both cell models. Together these data suggest a massive interaction of BOLD-100 with glycolysis and a stably enhanced glucose metabolization as a targetable resistance mechanism. The feasibility of this approach needs, for sure, confirmation in the in vivo situation. Animal experiments are planned in the near future.

Based on the synergy between BOLD-100 and glucose deprivation, we focused on additional molecular mechanisms known to be affected by both the ruthenium compound and 2-DG. GRP78 belongs to the HSP70 family and plays a major role as a chaperone for proper protein refolding due to the UPR and ER stress induction [79]. While GRP78 inhibition plays an important role in the mode of action of BOLD-100 [32,33,61], the ER stress-inducing activity of 2-DG is based on the prevention of the *N*-glycosylation of proteins, causing their improper folding and trafficking and leading to UPR induction [19,21,59,60]. We identified that HCTR cells display an overall decreased GRP78 expression as compared to parental cells, reflecting the GRP78 downregulating activity of BOLD-100, especially in response to protein damage stress [33]. As expected, 2-DG distinctly upregulated GRP78, based on the massive accumulation of proteins lacking proper *N*-glycosylation. In the parental cell line, BOLD-100 distinctly reduced GRP78 upregulation by 2-DG, suggesting that improper chaperone function and enhanced protein damage would be another factor underlying the synergism between the two compounds. However, GRP78 protein levels remained unchanged or even tended to be enhanced in HCTR cells under acute BOLD-100 treatment despite persisting ER stress activation, as proven by the phosphorylation of eIF2a. This suggests that acquired resistance is able to prevent GRP78-targeting by BOLD-100 despite persisting ER-stress induction.

The stronger ER stress-related response to the reduction of protein glycosylation in HCTR cells prompted us to investigate, in addition to the ER chaperone GRP78, also the cytosolic chaperone HSP70. Consistent with previous reports about a widely absent regulation of HSP70 expression upon BOLD-100 treatment [61], HSP70 levels remained comparably unchanged in HCT116 and HCTR cells in response to BOLD-100 treatment. However, unexpectedly, HSP70 levels were markedly reduced by 2-DG treatment in the resistant HCTR cells only. This selective decrease of HSP70 in cells with acquired BOLD-100 resistance indicates a generally disturbed proteostasis [65,80] induced by BOLD-100 selection and, thus, provides another explanation for the observed hypersensitivity of resistant cells towards 2-DG treatment. HSP70, as a central protein fate hub, is known to be implicated in several protein degradation pathways, including, amongst others, micro-, macro-, or chaperone-mediated autophagy [81]. Hence, the 2-DG-induced decrease of HSP70 expression in HCTR cells, together with the enhanced phosphorylation of eIF2A and the prominent cytoplasmic vacuole formation, links 2-DG treatment with a deregulation of autophagy [66].

Indeed, single agent 2-DG treatment in HCT116 and HCTR cells caused increased p62 and beclin-1 levels, suggesting enhanced autophagic activity, in accordance with the decreased lysosomal compartment. Importantly, the combination with CQ as an inhibitor of autophagosome-lysosome fusion potentiated the 2-DG-associated increase of LC3B I, indicating a buffering role of glucose in the induction of autophagy. In sharp contrast to 2-DG, treatment with BOLD-100 resulted in decreased p62 and beclin-1 levels, pointing to an autophagy-promoting activity of the compound. These results are supported by previous studies reporting the induction of autophagy by ruthenium-based chemotherapy in different cancer types [82,83,84]. However, the strong increase, not only in LC3B II but also LC3B I levels especially in combination with 2-DG, raises doubts about whether BOLD-100-induced autophagy is indeed functional. The increase in LC3B I, together with the reduction of cellular PE levels, suggests a defect in LC3B processing, potentially related to a defective lipidation by PE-conjugation. Interestingly, the accumulation of LC3B isoforms in response to BOLD-100 was absent in HCTR cells. However, in the combination setting with 2-DG, this effect was at least partly restored. These data strongly suggest that the enhanced glycolysis in HCTR cells plays a central role in the prevention of autophagy defects induced by BOLD-100. Additionally, the blockade of downstream autophagy by CQ also recovered, at least partly, the BOLD-100-induced increase of unprocessed LC3B I observed in parental cells. This suggests that lipid processing by acidic lysosomes might be essential for the resistance of HCTR cells against BOLD-100-induced LC3B conversion defects. The massive increase of both forms of LC3B in the combination of BOLD-100 with 2-DG in HCT116 cells, exceeding the effect of the ruthenium compound alone, indicates that glucose deprivation further promotes the BOLD-100-associated LC3 processing defect, hindering a functional autophagic processing. We conclude that HCTR cells survive treatment with the ruthenium compound via adaptation to BOLD-100-induced autophagy defects by re-established functional ER stress signaling. 2-DG interferes with this adaptation mechanism of HCTR cells, as seen by a distinct increase in eIF2a phosphorylation that leads to autophagy deregulation at the level of autophagosome-lysosome fusion. Targeting of this autophagy adaptation by 2-DG-induced glucose starvation re-sensitizes acquired resistant cells for BOLD-100 anticancer activity.

## 5. Conclusions

To summarize, this study reveals a significant glycolysis-blocking anti-Warburg effect as novel mechanism of action of BOLD-100. Metabolic rewiring contributes to acquired resistance development, translating into a targetable glucose dependency as a metabolic vulnerability. This metabolic deregulation in response to the ruthenium compound cross-talks with distinct proteostasis and autophagy defects. Whether these different but closely linked activities of BOLD-100 are the consequence of the interaction with one or several molecular targets is matter of ongoing investigations. In any case, inhibition of glycolysis represents a promising strategy to overcome acquired BOLD-100 resistance and enhance BOLD-100 anticancer activity. Finally, the detection of an upregulated glucose uptake associated with BOLD-100 exposure and acquired therapy resistance might be utilized clinically using ^18^F-fluoro-deoxy-glucose positron emission tomography (FDG-PET) [85]. Hence, we propose clinical studies on the use of ^18^F-FDG-PET to monitor both BOLD-100 anticancer response and resistance development.

## Figures and Tables

**Figure 1 pharmaceutics-14-00238-f001:**
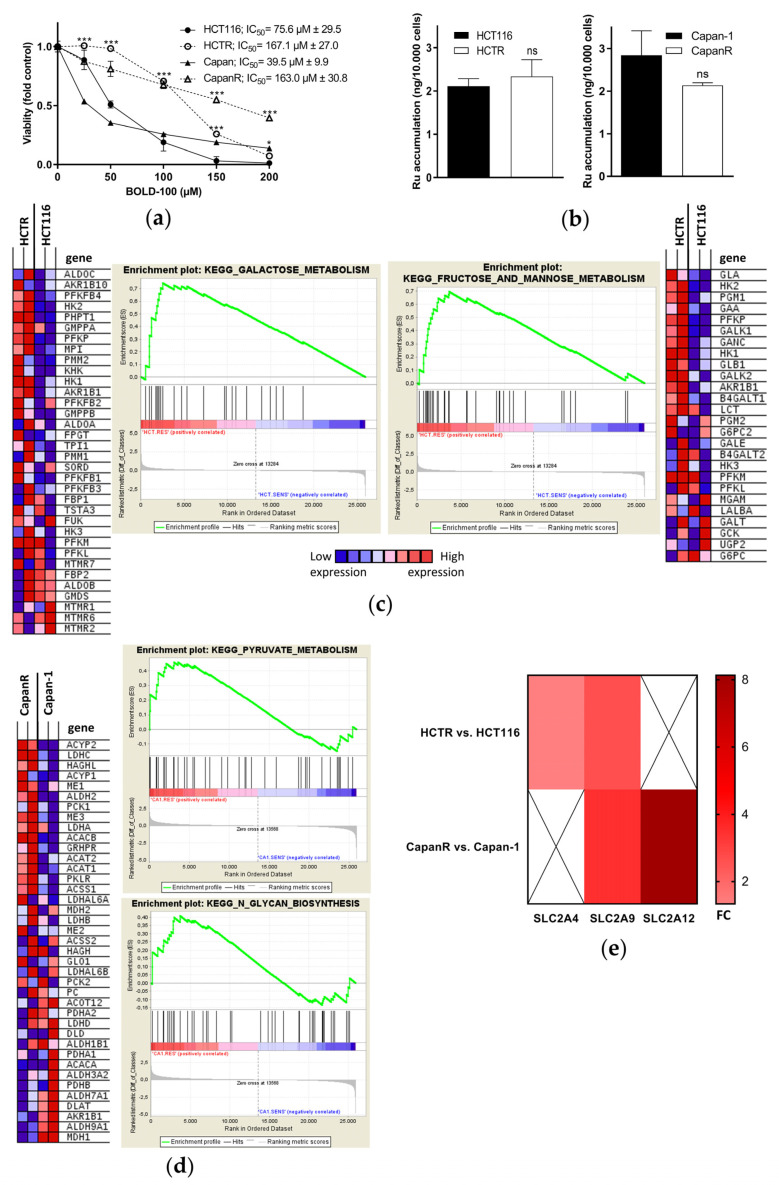
Whole genome gene expression profiling identifies transcriptional deregulations of cellular carbohydrate metabolism as a response to BOLD-100 and in acquired resistance against BOLD-100 in HCT116 and Capan-1 cells. (**a**) Cell viability of acquired BOLD100-resistant HCTR and CapanR cells as compared to their respective parental HCT116 and Capan-1 cells upon 72 h of treatment with increasing BOLD-100 concentrations, as determined by MTT assay. The statistical significance of the differences was determined using a two-way ANOVA with a Bonferroni post-test: * *p* < 0.05, *** *p* < 0.001. Asterisks are given above BOLD-100-resistant cells at the indicated concentrations and indicate the level of significance of a difference in comparison to parental cells. (**b**) ICP-MS analysis of whole cell lysates of indicated cells after 24 h of treatment with 100 µM BOLD-100. Statistical significance of differences was determined using a two-tailed unpaired Student´s *t*-test: no significant (ns) difference was detected. (**c**) GSEA of HCTR vs. HCT116 cells identifies “GALACTOSE_METABOLISM” (nominal *p*-value *<* E^−7^; FDR 0.010) and “FRUCTOSE_AND_MANNOSE_METABOLISM” (nominal *p*-value *<* E^−7^; FDR 0.084) as the third and fourth most significantly enriched gene sets in the KEGG database. Heat maps display the differentially regulated genes of the respective gene sets. (**d**) GSEA of CapanR vs. Capan-1 cells identifies “KEGG_PYRUVATE_METABOLISM” (nominal *p*-value 0.147; FDR 0.596) as the eighth and “KEGG_N_GLYCAN_BIOSYNTHESIS” (nominal *p*-value 0.193; FDR 0.703) as the tenth most significantly upregulated gene set. The heat map displays differentially regulated genes of the “KEGG_PYRUVATE_METABOLISM” gene sets. (**e**) mRNA regulation of SLC2A4, SLC2A9, and SLC2A12 of HCTR and CapanR as compared to respective parental counterparts. The bar on the right indicates fold-change (FC) values.

**Figure 2 pharmaceutics-14-00238-f002:**
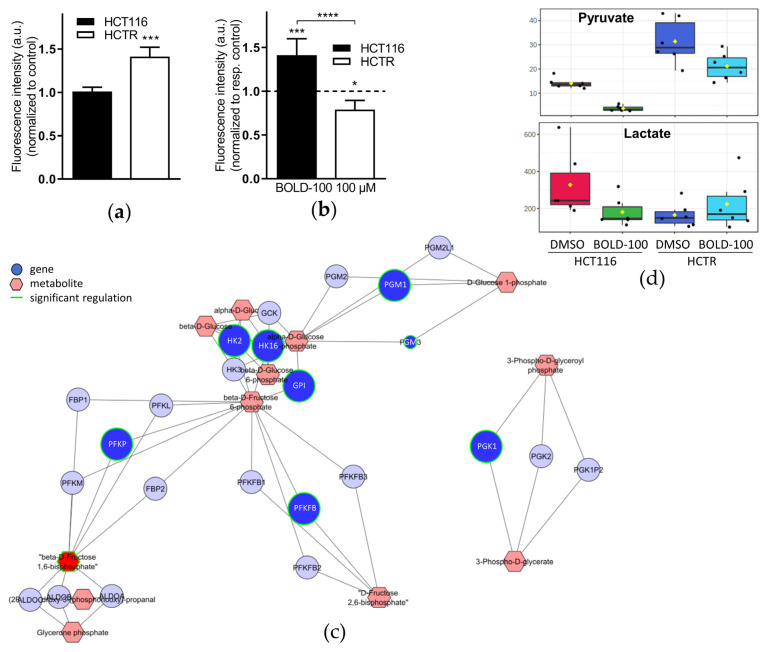
Acquired BOLD-100-resistant HCTR cells are characterized by enhanced glucose uptake and glycolytic activity, leading to increased intracellular pyruvate and decreased lactate levels. Flow cytometry analysis of fluorescence intensity of glucose-starved HCT116 and HCTR cells after 1 h of incubation with 25 µM 2-NBDG following treatment with solvent control (**a**) or 100 µM BOLD-100 (**b**) for 24 h. The results are the means of triplicates ± SD of two independent experiments. In (**b**) data are normalized to the respective control (dashed line). The statistical significance of differences was calculated with a two-tailed unpaired Student’s *t*-test (**a**) or a two-way ANOVA with Tukey’s multiple comparisons test (**b**): * *p* < 0.05, *** *p* < 0.001, **** *p* < 0.0001. (**c**) Gene-metabolite network of whole genome gene expression data computed together with metabolomics data of HCTR vs. HCT116 cells. Dots and hexagons indicate regulated genes or metabolites, respectively. Differences in the color intensity correspond to the strength of gene expression change and the green fringe indicates significant regulation. (**d**) Metabolomics of HCTR vs. HCT116 cells with 24 h of treatment with DMSO or 100 μM BOLD-100; all metabolites in pmol/μg protein, *n* = 6 biological replicates.

**Figure 3 pharmaceutics-14-00238-f003:**
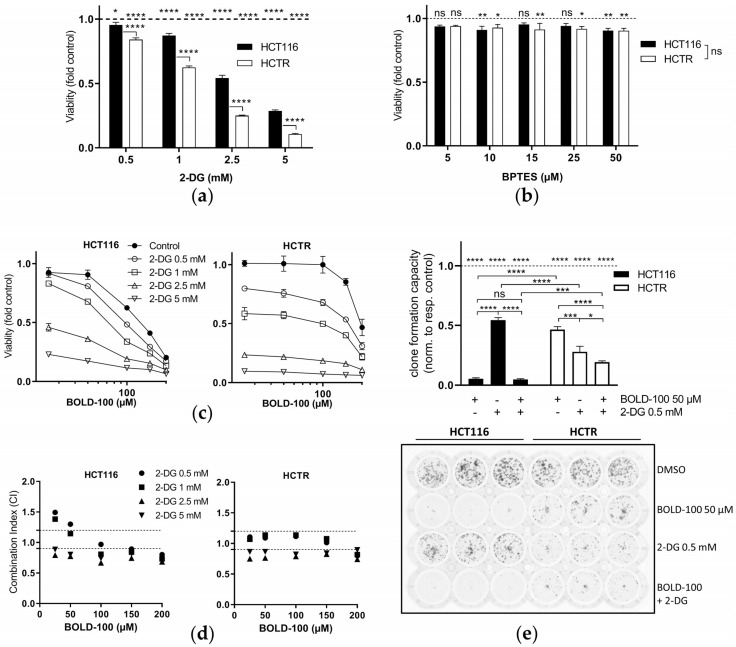
2-DG targets enhanced glycolytic activity in HCTR cells and synergizes with BOLD-100. (**a**) Cell viability assay of HCT116 and HCTR cells after 72 h of treatment with 2-DG relative to their respective controls (dashed line). Respective significance levels relative to the controls are indicated above the dashed line. The statistical significance of differences was determined using a two-way ANOVA with Tukey´s multiple comparisons test: * *p* < 0.05, **** *p* < 0.0001. The significance between respective treatment groups is given between the cell lines. (**b**) Cell viability assay of HCT116 vs. HCTR cells after 72 h of treatment with the glutaminase inhibitor BPTES relative to their respective controls (dashed line). The statistical significance of differences was determined using a two-way ANOVA with Tukey´s multiple comparisons test: * *p* < 0.05, ** *p* < 0.01; ns: non-significant. (**c**) Cell viability of indicated cells upon 72 h of treatment with BOLD-100 in combination with 2-DG at the indicated concentrations, as determined by MTT assay. One representative of three independent experiments is shown. (**d**) Combination indices (CI) based on the cell viability data from HCT116 and HCTR cells treated with BOLD-100 in combination with 2-DG at the indicated concentrations for 72 h. CI < 0.9, synergism; CI = 0.9–1.2, additive effects; or CI > 1.2, antagonism. (**e**) Crystal violet-stained clone formation assay showing clonogenic cell growth of HCT116 and HCTR cells treated with 50 µM BOLD-100, 0.5 mM 2-DG, or the combination of both compounds for seven days. For the solvent control, cells were treated with the amount of DMSO equivalent to the respective BOLD-100 samples. Quantification is shown in the lower panel. Results are normalized to the respective control (dashed line). The statistical significance of differences was determined with a two-way ANOVA with Tukey’s multiple comparisons test: * *p* < 0.05, *** *p* < 0.001, **** *p* < 0.0001.

**Figure 4 pharmaceutics-14-00238-f004:**
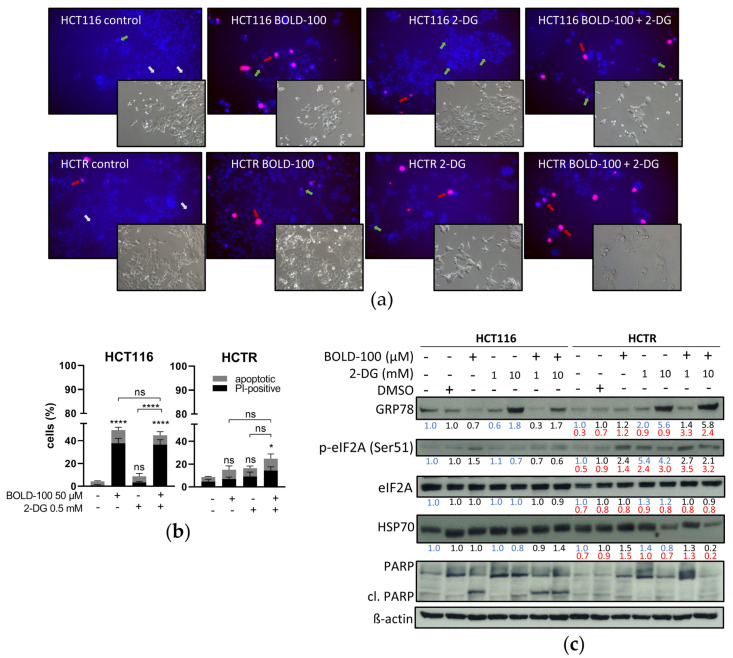
Glucose deprivation by 2-DG differentially regulates ER stress and leads to apoptotic cell death upon combination with BOLD-100. (**a**) Representative composite images show morphological changes of HCT116 or HCTR cells detected with dual staining of Hoechst 33342/PI. Cells were treated for 72 h with DMSO (control, equivalent to BOLD-100), 2.5 mM 2-DG, 100 µM BOLD-100 or their combination and imaged by fluorescence microscopy (magnification 20×). Grey arrows indicate examples of mitotic nuclei. Green arrows indicate live cells with apoptotic nuclei. Red arrows indicate dead cells with late apoptotic nuclei. (**b**) Quantification of treatment-respective early and late apoptotic nuclei depicted in (**a**). Statistical significance of differences was calculated using a two-way ANOVA with Tukey´s multiple comparisons test: * *p* < 0.05, **** *p* < 0.0001; ns: non-significant. (**c**) Expression levels of GRP78, peIF2A (Ser51), eIF2A, HSP70, and PARP after 24 h of treatment of HCT116 and HCTR cells with the indicated concentrations of BOLD-100, 2-DG, or their combination, analyzed by Western blotting. Two different control states were included, i.e., medium control without DMSO for 2-DG and with DMSO as a solvent control for BOLD-100. β-actin served as the loading control. The numbers below indicate quantified Western blot signal intensities normalized to their respective controls: blue, medium; black, DMSO; and red, HCTR vs. HCT116 cells.

**Figure 5 pharmaceutics-14-00238-f005:**
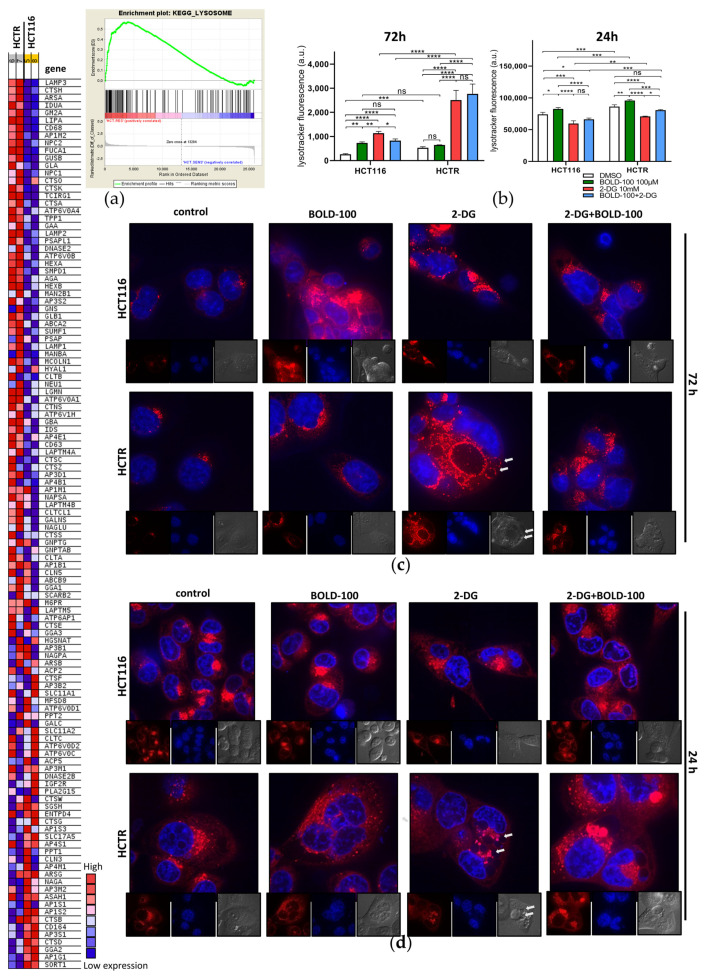
Survival and death from glucose deprivation by 2-DG is associated with differential regulation of the lysosomal compartment. (**a**) GSEA of HCTR vs. HCT116 cells identifies the “LYSOSOME” (nominal *p*-value < E^−7^; FDR 0.010) as the fifth most significantly enriched gene set in the KEGG database. The heat map displays differentially regulated genes of the respective gene set. (**b**) Flow cytometry analysis of fluorescence intensity of HCT116 and HCTR cells after 72 h or 24 h of treatment with DMSO, 100 µM BOLD-100, 10 mM 2-DG, or their combination, stained for 30 min with 0.5 µM Lysotracker red. The statistical significance of differences was calculated with a two-way ANOVA with Tukey´s multiple comparisons test: * *p* < 0.05, ** *p* < 0.01, *** *p* < 0.001, **** *p* < 0.0001; ns: non-significant. (**c**) Representative spinning-disc live cell fluorescence images (magnification 192×) of Lysotracker (red) and Hoechst 33342 (blue)-stained HCT116 and HCTR cells after 72 h or (**d**) 24 h of treatment with 100 µM BOLD-100, 10 mM 2-DG, or their combination.

**Figure 6 pharmaceutics-14-00238-f006:**
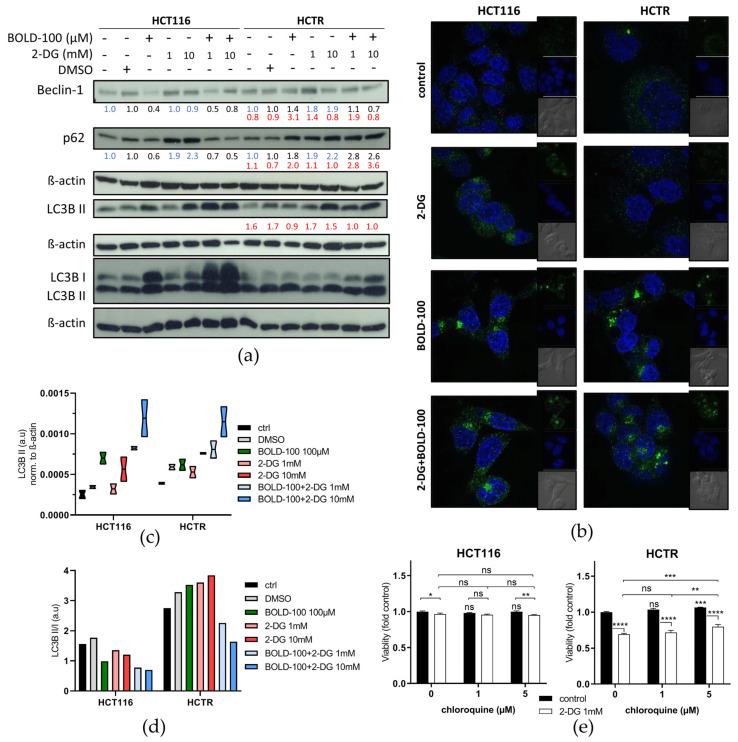
2-DG differentially regulates autophagy depending on BOLD-100 sensitivity or resistance, and the combination synergistically disturbs the autophagic flux. (**a**) Expression levels of beclin-1, p62, LC3B, and LC3B I/II in HCT116 and HCTR cells treated with the indicated concentrations of BOLD-100, 2-DG, or their combination for 24 h, analyzed by Western blotting. Two different control states were included, i.e., without DMSO for 2-DG and with DMSO as a solvent control for BOLD-100. β-actin served as the loading control. The numbers below indicate the quantified Western blot signal intensities normalized to their respective controls: blue, medium; black, DMSO; and red, HCTR vs. HCT116 cells. (**b**) Representative spinning-disc live cell fluorescence images (magnification 192×) of Lysotracker (red) and Hoechst 33342 (blue)-stained HCT116 and HCTR cells after 24 h of treatment with 100 µM BOLD-100, 10 mM 2-DG, or their combination. (**c**) Quantification of LC3B signal of two independent Western blot experiments. (**d**) Calculation of the treatment-respective LC3B II/I ratio from protein expression detected in (**a**). (**e**) Cell viability of HCT116 and HCTR cells upon 72 h of treatment with CQ in combination with 2-DG at the indicated concentrations, determined by MTT assay. The statistical significance of differences was calculated with a two-way ANOVA with Tukey´s multiple comparisons test: * *p* < 0.05, ** *p* < 0.01, *** *p* < 0.001, **** *p* < 0.0001; ns: non-significant.

## Data Availability

Not applicable.

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
