# Peer review of "The Anticancer Ruthenium Compound BOLD-100 Targets Glycolysis and Generates a Metabolic Vulnerability towards Glucose Deprivation"

_pharmaceutics, 2022, doi:10.3390/pharmaceutics14020238_

Round 1
Reviewer 1 Report
Comments:
This manuscript “The Anticancer Ruthenium Compound BOLD-100 Targets Glycolysis and Generates a Metabolic Vulnerability towards Glucose Deprivation” describes acquired resistance mechanism of BOLD-100. The introduction clearly discuss the purpose and backgrounds of the manuscript. The explanation and discussion are appropriate and clearly supported by the data. Overall, it needs a minor revision to be published.
- It would also very helpful to explain other ruthenium drugs under development or approved, comparing and considering BOLD-100.
- How long the BOLD-100-resistant sublines retain its drug-resistance?
- Figures should be re-constructed; For example, it is impossible to read any axis or name in the graph c and d. Also, it is complex to read which data is included in c and d. Statistic methods for each sub-figure should be noted in the figure legend.
- Please explain 2-DG vs. BPTES regarding Figure 3a,b in more detail.
- Is the combination index different statistically different in Figure 3d? In addition, CI seems to be not very impressive, considering numerous papers describing synergistic combination therapy.
- Is there any changes in the expression of immune checkpoints on drug-resistant cell lines compared to normal cancer cells?
Author Response
Reply to Reviewer 1
1. It would also very helpful to explain other ruthenium drugs under development or approved, comparing and considering BOLD-100.
Response: We appreciate this suggestion and included a paragraph about the two compounds NAMI-A and KP1019, entering clinical investigation before BOLD-100, in the introduction section from line 97 to 104. To the best of our knowledge, no ruthenium compound has been approved for clinical use so far.
2. How long the BOLD-100-resistant sublines retain its drug-resistance?
Response: This was tested by MTT assay after culturing BOLD-100-resistant cells for 1 month without the usual BOLD-100 selection. Parental, regularly (2 week intervals)-selected acquired resistant and 1 month-unselected acquired resistant cells were exposed for 72 h to BOLD-100. 1 month-unselected acquired resistant cells did not show reversal of BOLD-100 resistance as compared to the regularly selected acquired resistant cells as shown by comparable or even higher IC50 values. Hence, acquired BOLD-100 resistance is at least stable for 1 month after last regular drug selection. However, we would like to stress that tested cell models were selected for drug resistance every two weeks to maintain a stable selection process. Accordingly, this information has been added to the Materials and Methods section at line 136.
3. Figures should be re-constructed; For example, it is impossible to read any axis or name in the graph c and d. Also, it is complex to read which data is included in c and d. Statistic methods for each sub-figure should be noted in the figure legend.
Response: Thank you very much for this comment. Figure 1c and d have been re-adjusted accordingly which strongly enhances readability. Respective statistic methods have been added in the figure legend of each sub-figure.
4. Please explain 2-DG vs. BPTES regarding Figure 3a,b in more detail.
Response: Please refer to lines 59ff and 510 for brief explanations of the mode-of-action of 2-DG and BPTES, respectively. Besides glucose, glutamate is considered the second most important source to fuel the TCA cycle. BPTES inhibits glutamine synthetase, thus, depriving cells of glutamate. With the MTT assays in Figure 3a, b we show that HCTR cells are hypersensitive to only 2-DG and not BPTES as compared to parental cells. This proves that HCTR cell survival depends on a functional glucose but not glutamate metabolisation. Accordingly, a sentence has been added in the Results section at line 510.
5. Is the combination index different statistically different in Figure 3d? In addition, CI seems to be not very impressive, considering numerous papers describing synergistic combination therapy.
Response: We agree with the reviewer that the interaction is additive to moderately synergistic. However, as strong synergism (like e.g. in synthetically lethal combinations) is not expectable, considering that both BOLD-100 and 2-DG obviously interfere with the identical target, namely glycolysis. Additionally, in Figure 3d CI values of one representative MTT experiment out of three are shown, all delivering comparable CI values. However, estimation of significance is impossible here, as calculation of CI values for the solvent controls is not feasible. The key observation is here, that we could identify enhanced glycolysis dependency as a common denominator in two unrelated cell models with acquired BOLD-100 resistance. Resistant cells can be targeted by this feature using 2-DG and to an even higher extent in combination with BOLD-100.
6. Is there any changes in the expression of immune checkpoints on drug-resistant cell lines compared to normal cancer cells?
Response: Whole genome gene expression arrays were searched exemplarily for expression regulation of PDCD1 (PD-1) and CTLA-4 in HCTR vs. HCT116 or CapanR vs. Capan-1 cells and no changes were detected. However, GSEA analysis of HCTR vs. HCT116 cells identified “KEGG_ALLOGRAFT_REJECTION” (nominal p-value: 0.003, FDR: 0.014), “KEGG_AUTOIMMUNE_THYROID_DISEASE” (nominal p-value: 0.006, FDR: 0.035), “KEGG_HEMATOPOIETIC_CELL_LINEAGE” (nominal p-value: 0.002, FDR: 0.039), and “KEGG_COMPLEMENT_AND_COAGULATION_CASCADES” (nominal p-value: 0.002, FDR: 0.042) among the top 20 significantly changed gene sets from the KEGG database. Concerning the pancreatic cancer models, no significant changes of immune-related gene sets could be identified in CapanR vs Capan-1 cells.
Reviewer 2 Report
This manuscript reported the anticancer ruthenium compound BOLD-100 targets glycolysis and generates a metabolic vulnerability towards glucose deprivation, the strategy is novel, combining cellular metabolism with medicinal chemistry. The experiment design is complete and reasonable, however, the expression can be improved,
- The chemical structure of BOLD-100 should be shown, also the structure of 2-DG, they are very important.
- Fig. 1c, 1d and 5a showed the enriched genes, however, this picture is not clear, it’s too original.
- Fig. 2c showed the gene-metabolite network, the gene name in blue circle is not clear. Also the gene has different colors, what is the difference?
- Fig. 5, the heatmap in the left is very small, the information is not recognizable.
- The manuscript has no animal study, only using cell to reveal the mechanism, the limitation should be mentioned.
Author Response
Reply to Reviewer 2
1. The chemical structure of BOLD-100 should be shown, also the structure of 2-DG, they are very important.
Response: The chemical structures of BOLD-100 and 2-DG, created with SciFinder, are now included in the manuscript in Figure S1a, and referenced accordingly in the introduction.
2. Fig. 1c, 1d and 5a showed the enriched genes, however, this picture is not clear, it’s too original.
Response: Thank you very much for this comment. Figure 1c, d, and 5a have been re-adjusted for better readability.
3. Fig. 2c showed the gene-metabolite network, the gene name in blue circle is not clear. Also the gene has different colors, what is the difference?
Response: Thank you very much for pointing this out. The gene names in the dark blue circles have been changed to white color to increase the readability. Differences in the color intensity align with the strength of gene expression change. As an example, the strength of PFKP regulation is indicated by dark blue color and the significance of the change by the light green rim. Hence, light blue coloring indicates a minor, non-significant expression change. A sentence explaining the color intensity has been added to the legend of Figure 2c.
4. Fig. 5, the heatmap in the left is very small, the information is not recognizable.
Response: We strongly agree with the referee and the heatmap in Figure 5a has been enlarged to improve readability.
5. The manuscript has no animal study, only using cell to reveal the mechanism, the limitation should be mentioned.
Response: We fully agree with this argument of the referee. For sure, our in vitro findings need to be evaluated in the in vivo situation. Respective animal experiments are currently planned and will be conducted in the near future. Accordingly, a short paragraph concerning this limitation has been added to the discussion section in line 870.